# Effects of Music on the Quality of Life of Family Caregivers of Terminal Cancer Patients: A Randomised Controlled Trial

**DOI:** 10.3390/healthcare11141985

**Published:** 2023-07-09

**Authors:** Inmaculada Valero-Cantero, Cristina Casals, Milagrosa Espinar-Toledo, Francisco Javier Barón-López, Nuria García-Agua Soler, María Ángeles Vázquez-Sánchez

**Affiliations:** 1Puerta Blanca Clinical Management Unit, Malaga-Guadalhorce Health District, 29004 Malaga, Spain; inmaculada.valero.sspa@juntadeandalucia.es; 2ExPhy Research Group, Department of Physical Education, Instituto de Investigación e Innovación Biomédica de Cádiz (INiBICA), Universidad de Cádiz, 11519 Puerto Real, Spain; 3Rincón de la Victoria Clinical Management Unit, Malaga-Guadalhorce Health District, 29730 Malaga, Spain; milagrosa.espinar.sspa@juntadeandalucia.es; 4Department of Preventive Medicine, Public Health and Science History, Institute of Biomedical Research in Málaga (IBIMA), University of Malaga, 29007 Malaga, Spain; baron@uma.es; 5Department of Pharmacology, Faculty of Medicine, University of Malaga, 29010 Malaga, Spain; nuriags@uma.es; 6Department of Nursing, Faculty of Health Sciences, PASOS Research Group, UMA REDIAS Network of Law and Artificial Intelligence Applied to Health and Biotechnology, University of Malaga, 29071 Malaga, Spain; mavazquez@uma.es

**Keywords:** informal caregivers, complementary therapy, alternative therapy, palliative care, self-reported quality of life, self-perceived quality of life, client satisfaction, music medicine

## Abstract

The aim of this study was to investigate the effects of listening to self-chosen music on the quality of life of family caregivers of cancer patients receiving palliative home care. A total of 82 family caregivers were assigned either to the intervention group (*n* = 41) or to the control group (*n* = 41) in this double-blind, multicentre, randomised controlled clinical trial. The recruitment period was between July 2020 and September 2021. The intervention group received individualised pre-recorded music in daily 30 min sessions for 7 consecutive days. The control group was given a recorded repetition of the basic therapeutic training education also in 30 min sessions for 7 consecutive days. The primary endpoint assessed was the caregivers’ quality of life (Quality of Life Family Version and European Quality of Life visual analogue scale) before and after the intervention. The secondary endpoint was their perceived satisfaction with the intervention (Client Satisfaction Questionnaire). The music intervention was successful, producing a tangible improvement in the caregivers’ quality of life (*p* < 0.01) and satisfaction with the care provided (*p* = 0.002). The intervention was not only effective but produced no adverse effects. This study encourages the use of self-chosen music as a complementary intervention in nursing care for family caregivers of palliative cancer patients.

## 1. Introduction

Globally, cancer represents the second most common cause of death, with a staggering death toll of nearly ten million individuals in 2020 [1]. Prompt and effective palliative care can help manage physical symptoms as well as psychological and spiritual distress, improving patients’ quality of life and potentially prolonging survival. Patients with advanced cancer need palliative care as early as possible [2], which is typically administered at home, as it is often the preferred setting for patients to receive care while remaining in their social environment and while also minimizing the burden on hospital resources [3,4]. Therefore, home-based palliative care plays a critical role in comprehensive cancer care, providing patients with a supportive and comforting environment for receiving end-of-life care.

Home-based palliative care has demonstrated promising results in reducing hospital readmissions and healthcare costs. Despite the fact that palliative care is typically provided by healthcare professionals, including nurses, physicians, and social workers, who work together to manage patients’ symptoms and address their physical, emotional, and spiritual needs, home-based palliative care can be provided by health professionals but also relies on the essential contribution of one or more caregivers, usually family members, who provide social support and round-the-clock care with no financial compensation [5,6].

These functions—proving care and companionship for a person at the end of their life—often provoke major alterations in the caregiver, such as anxiety, fatigue, and an intense sensation of overload [7,8,9] together with depression [10], sleep disturbances [11], and impaired quality of life overall [12,13]. Supporting caregivers in the context of home-based palliative care is therefore crucial to ensure that both the patient and the caregiver receive the best possible care and support during this difficult time. Addressing caregivers’ well-being can contribute to more sustainable and successful home-based palliative care delivery. Therefore, it is essential to understand the unique challenges faced by caregivers in the context of home-based palliative care and to develop effective interventions to support them.

In this regard, the significance of addressing the quality of life of family caregivers of patients with cancer in palliative care is underscored. The parameter “quality of life” is composed of different domains reflecting an individual’s general level of health and well-being. It can be measured using dimensions with physical, psychological, social, and spiritual components [14]. For caregivers of patients in palliative care, their quality of life is often reduced by alterations such as overload [15] and the absence of necessary support [13]. In accordance with the indications of the World Health Organization, palliative care is provided to alleviate symptoms and to enhance the patient’s quality of life [16]. During home-based palliative care, interventions should also be conducted to assure caregivers’ own quality of life, helping them provide optimum levels of care [17,18].

One such intervention may be that of psychoeducation [19], which refers to any structured psychological educational program. However, despite the acknowledged importance of assuring caregivers’ quality of life, few studies have been conducted to determine the effectiveness of this and related interventions [17]. One example is music medicine, which is defined as “when medical or health personnel offer pre-recorded music for passive listening” [20]. This is a complementary musical intervention, which is defined as “a group of diverse medical and health care systems, practices, and products that are not generally considered part of conventional medicine” [21]. Regarding listening to music, auditory signals are an important aspect of human sensory systems. Among them, music has a special capacity that is capable of inspiring emotions. The temporal limbic system, formed, among other structures, by the amygdala, the hippocampus, the ventral and dorsal striata, and the auditory cortex, helps decode the emotional value of the music heard [22].

Music may both create emotions and alter moods [22,23]. For this reason, listening to music is sometimes used as a complement in the care of patients with cancer to improve fatigue, depression, and sleep and decrease pain, among other symptoms [24,25,26]. However, this benefit has not been demonstrated in all the symptoms studied, and therefore, more studies are required in this regard [27]. Focusing on caregivers of patients with advanced cancer, it has been previously shown that music has benefits on symptoms of anxiety, fatigue, depression, and blood volume pulse amplitude [28,29]. However, despite the importance of quality of life as a health indicator, to our knowledge, no previous clinical trials have been performed to evaluate the advantages of music for family caregivers of palliative-care patients in the home [30].

The aim of the present study is to investigate the potential benefits of a complementary music medicine intervention for the family caregivers of patients in palliative home care. Our main study hypothesis was that the intervention group would achieve a greater improvement in quality of life than the control group after the seven-day intervention. The secondary hypothesis was that the intervention group would also report greater satisfaction than the control group with the therapy received.

## 2. Materials and Methods

### 2.1. Design

To address the previous hypotheses, we conducted a double-blind, multicentre, randomised controlled clinical trial of family caregivers for cancer patients in palliative home care [31]. The intervention group received a complementary music medicine intervention for seven consecutive days, while the control group received conventional treatment also for seven consecutive days. Both groups were assessed for quality of life and satisfaction with therapy at baseline and after the intervention period. The study is registered at Clinical Trials.gov, reference code: NCT04052074; registered on 9 August 2019. 

We recently published a scientific article of this short-term trial [32], suggesting that using self-chosen music significantly reduced caregiver burden among family caregivers of patients with advanced cancer. While burden focuses on the negative aspects of caregiving, such as the level of strain, stress, and disruptions to the caregiver’s life, quality of life provides a broader perspective that includes positive aspects, such as personal fulfilment, social support, and the caregiver’s ability to find meaning in their caregiving role. Burden is often related to the objective demands of caregiving, such as the intensity of care required, financial strain, and time commitments. Quality of life, on the other hand, reflects the caregiver’s subjective appraisal of their well-being, encompassing physical health, emotional well-being, social functioning, and overall life satisfaction.

Therefore, the present manuscript specifically presents the quality-of-life results of this clinical trial since it allows for a clearer and more focused presentation of findings. This level of depth and granularity enhances the scientific understanding of each variable and contributes to the overall body of knowledge in the field.

### 2.2. Participants

The study participants were all family caregivers for cancer patients in palliative home care, recruited at six primary-care clinical management units within the Malaga-Guadalhorce Health District from Andalusia, Spain. The participants were recruited by reference to the lists of patients included in the Palliative Care Assistance Process in the DIRAYA Digital Health Record maintained at each of the above units, using convenience sampling. The following inclusion criteria were applied: (i) family caregiver; (ii) age at least 18 years. Any caregivers with severe hearing loss preventing them from using an mp3 device or mobile phone were excluded from the study group.

### 2.3. Research Ethics

This clinical trial underwent a comprehensive ethical review process and was granted approval by the Research Ethics Committee of the Province of Malaga on 28 March 2019 (reference code: AP-0157-2018). The study adhered to the principles outlined in the Declaration of Helsinki, which ensures the protection of participants’ rights, welfare, and privacy during research studies. All participants were provided with detailed information regarding the objectives, procedures, and potential risks and benefits of the study, both verbally and in writing, and gave prior signed informed consent. 

### 2.4. Measures

The main outcome was the caregivers’ quality of life, which was assessed using the following scales:-The Quality of Life Family Version (QOL-FV) 14, validated in Spanish [33], which measures the quality of life of family caregivers for patients with cancer, consists of thirty-seven items, scored on a scale ranging from 0 = worst result to 10 = best result (although several are coded by means of an inverse score). The instrument has four sub-scales or domains: physical well-being, psychological well-being, social concerns, and spiritual well-being;-The European Quality of Life—5 dimensions (EuroQol-5D-5L) questionnaire was developed in the United Kingdom and in Spain [34]. The questionnaire score is obtained using a visual analogue scale, ranging from 0 to 100, representing “worst imaginable state of health” to “best imaginable state of health”, respectively.

Moreover, the secondary outcome was the caregivers’ satisfaction with the intervention, and it was assessed using the Client Satisfaction Questionnaire (CSQ-8) [35], validated in Spanish [36]. This self-administered questionnaire consists of eight questions scored on a 4-point Likert-type scale (in four cases, the score is inverse). The final assessment is obtained by summing these scores and can range from 8 to 32 points.

### 2.5. Sample Size and Randomisation

The necessary sample size was calculated according to the method described by Hanser et al. [37], assuming a standard deviation of 1.23 for satisfaction, a power of 80%, and an alpha error of 5%. Furthermore, an increase of one point in caregiver satisfaction was considered relevant. Under these premises, the study required two samples of 34 caregivers. To allow for a possible dropout rate of 20%, this sample size was increased to two samples of 41 caregivers.

The randomisation of the participants between the control and intervention groups was carried out as follows. Cards were marked “Intervention group” or “Control group” and inserted into sealed opaque envelopes, which were then shuffled and numbered. Each participant was assigned a number by order of enrolment. The researcher performing the randomisation then opened the envelope corresponding to each number in turn and assigned the participant accordingly. In this double-blind trial, therefore, neither the caregivers nor the evaluators knew the group to which each participant belonged.

### 2.6. Intervention, Control Group, and Masking

All participants received conventional health care before the intervention in same conditions with basic therapeutic education for palliative care, which is part of the regular health care service in our health centres and is provided by nurses.

Then, the caregivers in the intervention group received, apart from the mentioned conventional health care, a music medicine intervention using pre-recorded music provided via an mp3 device or mobile phone. The music used was individualised, chosen by each participant to ensure that the music enhanced their well-being. The intervention consisted of a 30 min music session received daily for seven days. The caregiver was instructed not to perform any other activity during the session.

The participants in the control group received the conventional health care, and in addition, in order to mask the fact of their belonging to the control group (both to the participants and to the evaluators), they received a recorded repetition of this conventional treatment using an mp3 device or mobile phone. This activity was also performed in 30 min sessions received daily for seven consecutive days. 

All study participants and clinical trial evaluators were blinded to the participants’ membership of the control or the intervention group. The study assessments were carried out by the case-manager nurses of the respective primary-care clinical management units.

### 2.7. Data Collection and Analyses

The study data were compiled between July 2020 and September 2021 at interviews during home visits and via self-administered questionnaires. If any caregiver had difficulty understanding or responding to the questionnaire items, they were helped by the corresponding case-management nurse. Data on the participants’ quality of life for both the control and the intervention groups were assessed before the intervention and seven days later after its conclusion. Their satisfaction was also assessed at this second time point.

The primary outcomes considered were the changes in the QOL-FV and EuroQol-5D-5L scores, between the baseline and the endpoint, and the outcomes achieved by the members of the control group with respect to those in the intervention group.

The secondary outcome considered was the participants’ satisfaction with the intervention as assessed using CSQ-8 for the control group in comparison with the intervention group.

The inter-group comparisons of the questionnaire scores obtained were made using the non-parametric Mann–Whitney test. Moreover, the groups were separately analysed comparing the pre-post differences with respect to 0 by applying a one-sample Wilcoxon test. The quantitative variables are expressed as medians, means, and standard deviations. A two-sided *p*-value < 0.05 was considered statistically significant. All data analyses were performed using SPSS version 23.0 statistical software on an intention-to-treat basis.

## 3. Results

Results were obtained from all 82 participants. No adverse events were reported, and all participants completed the full programme. The flow diagram for the study is presented in Figure 1. Regarding baseline characteristics, there were no significant differences between the two groups (Table 1). The following change-from-baseline results were obtained: intervention vs. control group for the EuroQol-5D-5L: 6.29 (10.71)/−3.76 (7.26), *p* < 0.001; total QOL-FV: 0.16 (0.75)/−0.35 (0.94), *p* = 0.008; CSQ-8 final score: 27.54 (3.25)/24.80 (4.03), *p* = 0.002 (see Table 2).

## 4. Discussion

This clinical trial was undertaken to investigate the benefits of music in relation to the quality of life reported by family caregivers of cancer patients in palliative home care and to determine these caregivers’ satisfaction with the health care provided. The study results reveal a significant improvement in the quality of life reported and increased satisfaction with the health care provided.

The caregivers of palliative-care patients normally experience a worsening quality of life as the disease progresses [38], even in 7 days [39]. Although a previous study found no significant difference in quality of life for caregivers who listened to music [28], our clinical trial did reveal such an improvement according to the total QOL-FV scores obtained by the intervention group. This difference may be due to the fact that the previous study was quasi-experimental [28]. Moreover, the music used was that recommended by the authors, while our clinical trial used the caregivers’ own choices of music.

Our analysis of the results for the four QOL-FV subscales reflect improved physical well-being and social concerns following the music intervention. However, there were no significant differences for the spiritual scale, indicating that although music medicine seems to enhance the quality of life in general, it does not have this effect regarding spiritual/religious aspects. A previous study conducted on another continent highlighted spiritual support as a need for family caregivers of cancer patients [40]. This finding may be related to the participants’ faith (or lack of it) and/or its possible loss due to the disease process experienced [41]. More studies are encouraged, as a review article concluded that studies of psychosocial interventions aimed at improving the quality of life of both cancer patients and their family caregivers pay relatively little attention to spiritual well-being [42]. In addition, although the intervention group reported improved psychological well-being in our study, there was no significant difference in this respect compared to the control group.

The above findings, referring to the results obtained with the QOL-FV scale, were corroborated by the evaluation based on the EuroQol-5D-5L scale, which, among other items, concerns the “quality of life today”. The congruent results obtained through the utilization of two distinct scales provide support for our research hypothesis, which posits that the implementation of self-chosen music interventions can enhance the quality of life experienced by family caregivers of cancer patients undergoing palliative care in a home setting.

In addition to addressing the study hypotheses, the potential practical applications of our findings are significant. The positive results seen in the intervention group highlight the potential for complementary music interventions to be used as a valuable tool in improving the quality of life and satisfaction of family caregivers in the context of home-based palliative care. The cancer-survivorship care plan should thoroughly acknowledge the intricate and enduring impact of unaddressed caregiver needs on their distinct dimensions of quality of life [43]. This could ultimately lead to better experiences for both caregivers and patients while reducing the burden on healthcare resources. Thus, the potential for widespread adoption and implementation in clinical settings is encouraged.

Indeed, we previously proposed this complementary music intervention as a useful strategy to reduce caregiver burden among these participants, suggesting its potential as a short-term relief strategy in family caregivers of palliative-care cancer patients [32]. However, the relationship between burden and quality of life is not consistent. As is shown in Table A1, the caregivers’ burden was statistically associated only with the VAS of the EuroQol-5D-5L (moderate relationship), without significant associations with any subscale or the total score of the QOL-FV. The absence of a statistically significant relationship between the caregivers’ burden and their quality of life is due to the fact that both concepts differ [44]. On the one hand, caregiver burden can be defined as the distress that caregivers feel as a result of providing care due to their caregiving responsibilities. It encompasses various aspects, such as the demands of providing care, financial difficulties, disruption of daily routines, and emotional distress. On the other hand, quality of life refers to the subjective well-being and satisfaction in various domains of life, including physical, psychological, social, and spiritual aspects [44].

Therefore, interventions aimed at supporting family caregivers should not only address the reduction of caregiver burden but also focus on improving their quality of life. These interventions may include self-chosen music sessions [45]. In the context of family caregivers of cancer patients, maintaining and enhancing their quality of life is of paramount importance to ensure their well-being and ability to continue providing effective care. Our study adds to the growing body of literature on the effectiveness of complementary therapies in healthcare, particularly in the context of palliative care, and highlights the potential for further research and development of such interventions. The practical applications of our findings underscore the importance of incorporating complementary therapies into traditional nursing care, particularly for those caring for terminally ill patients at home, in order to provide comprehensive and holistic support to their caregivers.

Another aspect studied is that of the caregivers’ satisfaction with the care received. This question is important because it is not sufficient for a novel intervention to produce an objective improvement; it must also be considered appropriate, agreeable, and useful by those for whom it is intended since the success of any such intervention depends on the participants’ cooperation and adherence. There are many determinants of satisfaction with health care, but the positive signs indicated in our study suggest that the musical medicine intervention was beneficial to the participants’ physical and mental health [46]. This conclusion is corroborated by the fact that the caregivers in the intervention group were significantly more satisfied with the care received than those in the control group.

These findings highlight the potential for complementary musical interventions to enhance overall patient and caregiver experiences in the context of home-based palliative care, ultimately leading to improved quality of life and outcomes for those receiving care. Further research is necessary to explore the full range of potential applications and benefits, ultimately paving the way for the integration of these interventions into standard care practices and ensuring the delivery of comprehensive palliative care as well as in different settings and populations. 

While the results of our clinical trial have provided novel and promising findings, it is important to acknowledge certain limitations that may affect the generalizability and long-term implications of our findings. First, one limitation is that our study sample consisted of participants recruited solely from public health clinics. Moreover, they were all located in urban areas. For both of these reasons, our findings may not reflect the experiences and outcomes of individuals utilizing private healthcare systems or those residing in rural areas, and therefore, caution should be exercised when generalising our results to broader populations. Consequently, the generalisability of our results may be limited. To address this limitation, future research should aim to investigate the effectiveness of complementary musical interventions in diverse populations, including individuals from various cultural backgrounds. By examining the impact of these interventions across different settings and demographics, a deeper comprehensive understanding of the potential impact of these interventions and the development of tailored approaches to meet the specific needs of different populations could be achieved.

Second, an aspect to consider is the durability of the effects observed in our study. It has not been established whether the effectiveness of music medicine persists over time or whether other similar interventions employed on a regular basis would be equally useful. Furthermore, the control group in this study underwent a repetition of the standard treatment consisting of varying 30 min sessions conducted over a 7-day period. Notably, participants were informed in advance by a nurse about this arrangement. This blinded design was implemented to address internal validity concerns, as it is known that receiving additional attention or care can positively influence quality of life and satisfaction with received care. However, it is crucial to acknowledge that the recorded conventional treatment itself may have influenced participants’ quality of life. Consequently, future studies should consider comparing different conditions and groups to obtain a more comprehensive understanding of the topic. Therefore, further studies could compare different conditions and groups to provide a more comprehensive understanding. Nevertheless, the method we describe was applied very rigorously, and it can be replicated without difficulty.

Finally, a notable constraint of this study lies in the temporal scope of the intervention, which is confined to a mere 7-day period. Further research is needed to investigate the long-term effectiveness of self-chosen music sessions and to compare it with other similar interventions that could be integrated into routine care.

## 5. Conclusions

The intervention group exhibited a significant improvement in quality of life compared to the control group, indicating that the intervention positively impacted their overall well-being. Additionally, the intervention group reported higher levels of satisfaction with the therapy received, suggesting a positive subjective experience. In conclusion, the results of our study provide evidence to support the use of complementary musical interventions in the context of home-based palliative care for family caregivers of terminally ill patients.

Our intervention was shown to be highly effective, producing significant improvements in both the quality of life and satisfaction with care of participants without any reported adverse effects. The positive outcomes seen in our study highlight the potential benefits of incorporating complementary therapies into traditional nursing care, particularly for those caring for terminally ill patients at home. By incorporating music as a complementary intervention, we can address their emotional, psychological, spiritual, and social well-being.

Future research is needed to further explore the potential applications and benefits of such interventions in different populations and settings. In addition, it is encouraged to thoroughly analyse the potential long-term effects of such interventions to ensure their sustained efficacy in the context of home-based palliative care and also to corroborate whether the implementation of complementary musical interventions in nursing care could ultimately lead to improved outcomes and experiences for both patients and their caregivers.

## Figures and Tables

**Figure 1 healthcare-11-01985-f001:**
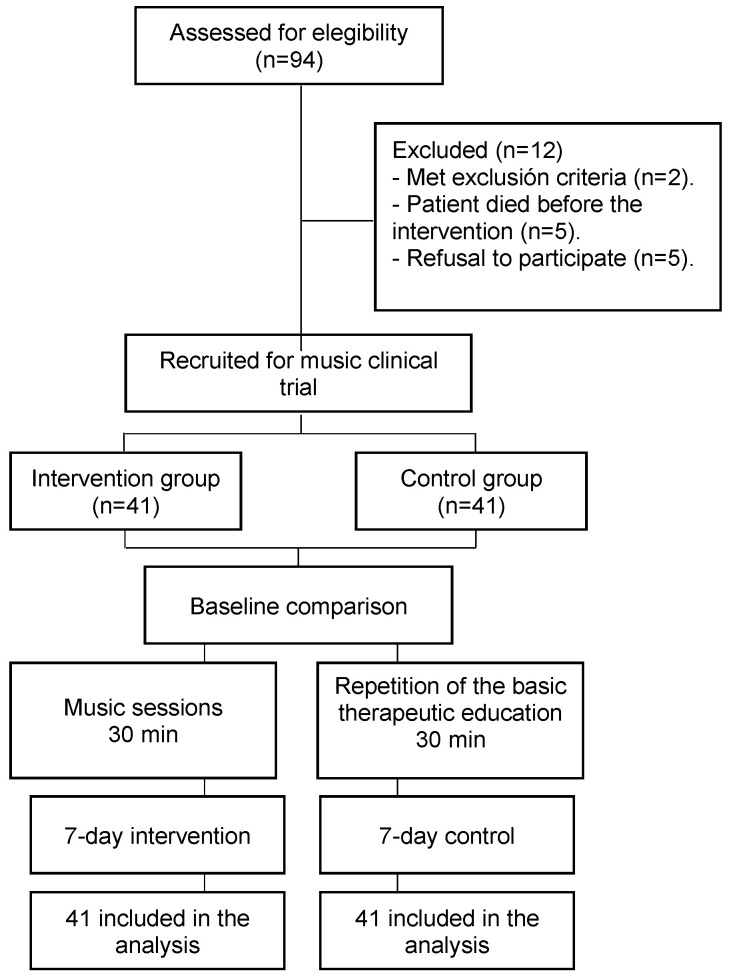
Flowchart of the study.

**Table 1 healthcare-11-01985-t001:** Socio-demographic characteristics of caregivers.

	Total (*n* = 82)	Intervention Group (*n* = 41)	Control Group (*n* = 41)	*p*-Value
Age (years)	62.71 (12.56)	63.12 (12.81)	62.31 (12.46)	0.626
Sex				
Female	72 (87.8%)	34 (82.9%)	38 (92.7%)	0.177
Male	10 (12.2%)	7 (17.1%)	3 (7.3%)	
Education				
No formal education	11 (13.4%)	5 (12.2%)	6 (14.6%)	1.000
Primary	39 (47.6%)	21 (51.2%)	18 (43.9%)	
Secondary	22 (26.8%)	10 (24.4%)	12 (29.3%)	
University	10 (12.2%)	5 (12.2%)	5 (9.8%)	
Relationship to the person cared for				
Spouse	47 (57.3%)	27 (65.9%)	20 (48.8%)	0.101
Daughter/son	25 (30.5%)	8 (19.5%)	17 (41.5%)	
Other	10 (12.2%)	6 (14.6%)	4 (9.8%)	
Hours of daily care	17.47 (7.14)	18.07 (6.98).	16.87 (7.34)	0.448
VAS of the EuroQol-5D-5L	82.61 (14.79)	81.71 (15.99)	83.51 (13.63)	0.694
QOL-FV (scores)				
Physical well-being	7.08 (2.46)	7.12 (2.42)	7.04 (2.53)	0.948
Psychological well-being	5.56 (1.51)	5.42 (1.62)	5.71 (1.38)	0.649
Social concerns	5.95 (1.93)	5.91 (2.04)	6.00 (1.83)	0.981
Spiritual wellness	3,98 (1,81)	4.17 (1.85)	3.79 (1.78)	0.452
Total QOL-FV	5.64 (1.43)	5.65 (1.58)	5.63 (1.28)	0.742

Results are presented as mean (standard deviation) or numbers (%). *p*-value corresponds to the Mann–Whitney U-test, except for sex (chi-squared test) and education (Fisher’s exact test). VAS of the EuroQol-5D-5L, visual analogue scale of the European Quality of Life—5 dimensions; QOL-FV, Quality of Life Family Version.

**Table 2 healthcare-11-01985-t002:** Outcomes differences between groups before and after the 7-day intervention.

		Intervention Group				Control Group			*p*-Value between Groups
	Pre	Post	Change	*p*-Value	Pre	Post	Change	*p*-Value	
QOL-FV (scores)									
Physical well-being	7.12 (2.47)	7.56 (1.84)	0.44 (1.70)	**0.006**	7.04 (2.53)	6.18 (2.34)	−0.86 (1.89)	0.106	**0.004**
Psychological well-being	5.42 (1.67)	5.63 (1.58)	0.22 (1.00)	0.297	5.71 (1.39)	5.53 (1.60)	−0.18 (1.09)	0.272	0.119
Social concerns	5.91 (2.04)	6.19 (1.84)	0.28 (1.03)	0.233	6.00 (1.83)	5.69 (2.04)	−0.31 (1.62)	0.065	**0.024**
Spiritual wellness	4.17 (1.85)	3.97 (1.54)	−0.30 (1.27)	0.825	3.79 (1.78)	3.76 (1.85)	−0.03 (1.00)	0.196	0.883
Total QOL-FV	5.65 (1.58)	5.81 (1.23)	0.16 (0.75)	**0.023**	5.63 (1.28)	5.28 (1.25)	−0.35 (0.94)	0.102	**0.008**
VAS of the EuroQol-5D-5L	81.71 (15.99)	88.00(13.03)	6.29 (10.71)	**0.002**	83.51 (13.63)	79.76 (12.1)	−3.76 (7.26)	**0.001**	**<0.001**
CSQ-8 (score)	-	27.54 (3.25)	-	-	-	24.80 (4.03)	-	-	**0.002**

*p*-value corresponds to pre-post differences respect to 0 in the Wilcoxon test, while *p*-value between groups corresponds to control group change versus that of the intervention group in the Mann–Whitney U-test. All *p*-values < 0.05 are presented in bold. VAS, visual analogue scale; CSQ-8, Client Satisfaction Questionnaire.

## Data Availability

Study data are available on reasonable request to the corresponding author.

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
