# Peer review of "Effects of Music on the Quality of Life of Family Caregivers of Terminal Cancer Patients: A Randomised Controlled Trial"

_healthcare, 2023, doi:10.3390/healthcare11141985_

Round 1
Reviewer 1 Report
Effects of music on the quality of life of family caregivers of 2 terminal cancer patients: a randomised controlled trial
Interesting research! The concept of music has been “studied” anecdotally, but this adds to the evidence of an easy intervention to be employed or encouraged by nurses. It seems that this paper has been revised from a previous review, therefore a few comments to consider. The paper as presented is well structured and the research process flows nicely now (as evidenced by the red updates).
Abstract line 26 control group give recorded repetition of basic therapeutic training, but in design, line 111 statement that control group received standard of care without any additional intervention. This is repeated in line 181 and described in 191. Provide a consistent description throughout these points in the paper. Section 2.6 describes it very well. But essentially the control group is receiving education twice (which is an additional intervention, although not unique as compared to music medicine). Suggest repeating wording to be consistent such as ‘conventional treatment’ rather than standard
Pg 1, line 46 change to “their social environment, while also minimizing the burden”
Line 80 music medicine is brought up for the first time here and the term is unique. Quite choppy transition from psychoeducation to music medicine. Suggest something like, “One example is music medicine, which is defined as…” Presuming the psychoeducation definition refers to the conventional treatment delivery? Could this be somehow added here to add consistency?
Line 306 not a new paragraph
Line 307 this is confusing. Public health clinics are not primary health clinics? Suggest public primary health clinics as the term and delete the second half of the sentence
Line 339 curious at the mention of only three domains, but no statement on future research into spirituality?
none
Author Response
Interesting research! The concept of music has been “studied” anecdotally, but this adds to the evidence of an easy intervention to be employed or encouraged by nurses. It seems that this paper has been revised from a previous review, therefore a few comments to consider. The paper as presented is well structured and the research process flows nicely now (as evidenced by the red updates).
Point 1: Abstract line 26 control group give recorded repetition of basic therapeutic training, but in design, line 111 statement that control group received standard of care without any additional intervention. This is repeated in line 181 and described in 191. Provide a consistent description throughout these points in the paper. Section 2.6 describes it very well. But essentially the control group is receiving education twice (which is an additional intervention, although not unique as compared to music medicine). Suggest repeating wording to be consistent such as ‘conventional treatment’ rather than standard.
Response 1: Corrected after suggestion. Thank you very much for your detailed review.
Point 2: Pg 1, line 46 change to “their social environment, while also minimizing the burden”.
Response 2: It has been modified according to the reviewer’s comment.
Point 3: Line 80 music medicine is brought up for the first time here and the term is unique. Quite choppy transition from psychoeducation to music medicine. Suggest something like, “One example is music medicine, which is defined as…” Presuming the psychoeducation definition refers to the conventional treatment delivery? Could this be somehow added here to add consistency?
Response 3: The sentence has been included as suggested. Also, the psychoeducation defition has been included to add clarity, but usually psychoeducation is not considered a conventional treatment.
Point 4: Line 306 not a new paragraph.
Response 4: Done.
Point 5: Line 307 this is confusing. Public health clinics are not primary health clinics? Suggest public primary health clinics as the term and delete the second half of the sentence.
Response 5: We agree, it has been corrected after suggestion.
Point 6: Line 339 curious at the mention of only three domains, but no statement on future research into spirituality?
Response 6: We also agree so it has been included. Thank you very much for all your comments, we really apprecite them.

Reviewer 2 Report
This is a very interesting study. The theme is current and contributes to a better scientific understanding of the phenomenon under study. The Introduction integrates the approach of the concepts under study, the objective and the hypotheses to be tested. It emphasizes the objective of the study. It adequately describes the methodology used. The results are clear and respond objectively and adequately to the central questions of the study. The discussion can be further enhanced with the use of more authors and studies that give it more scientific solidity. The limitations of the study are presented with logic and methodological consistency. Conclusions are adequate.
Author Response
Point 1: This is a very interesting study. The theme is current and contributes to a better scientific understanding of the phenomenon under study. The Introduction integrates the approach of the concepts under study, the objective and the hypotheses to be tested. It emphasizes the objective of the study. It adequately describes the methodology used. The results are clear and respond objectively and adequately to the central questions of the study. The discussion can be further enhanced with the use of more authors and studies that give it more scientific solidity. The limitations of the study are presented with logic and methodological consistency. Conclusions are adequate.
Response 1: Thank you very much for your comments. We greatly appreciate your feedback. We have made efforts to include more studies in the discussion depite the limited availability of research on this specific topic. Thanks.
Reviewer 3 Report
This research investigates the effects of music on the quality of life of family caregivers in the aspects of physical well-being, psychological well-being, social concerns, spiritual wellness, and overall QoL. While it is important to find ways for caregivers to improve their QoL, the proposed method has potential flaws that keep the manuscript from publishable in the journal.
The participants were randomly assigned to a control group and a treatment group. The treatment is a 30 min music session for 7 consecutive days. The participants in the control group, on the other hand, had to join the repetitive 30 min of basic therapeutic education for 7 consecutive days. My view to this study design is not to investigate the effect of music, but to test how much the negative effect of boredom in therapeutic education could be. Table 2 reveals the effects of these two treatments, music treatment and treatment of boredom from repetitive education. In 3 out of 5 QoL items, the negative effect of boredom outweighs the positive effect of music.
In conclusion, the control method of this study is poorly designed.
English language quality is fine.
Author Response
Point 1: This research investigates the effects of music on the quality of life of family caregivers in the aspects of physical well-being, psychological well-being, social concerns, spiritual wellness, and overall QoL. While it is important to find ways for caregivers to improve their QoL, the proposed method has potential flaws that keep the manuscript from publishable in the journal.
The participants were randomly assigned to a control group and a treatment group. The treatment is a 30 min music session for 7 consecutive days. The participants in the control group, on the other hand, had to join the repetitive 30 min of basic therapeutic education for 7 consecutive days. My view to this study design is not to investigate the effect of music, but to test how much the negative effect of boredom in therapeutic education could be. Table 2 reveals the effects of these two treatments, music treatment and treatment of boredom from repetitive education. In 3 out of 5 QoL items, the negative effect of boredom outweighs the positive effect of music.
In conclusion, the control method of this study is poorly designed.
Response 1: Thank you for your feedback on our research manuscript. We would like to address the concerns raised regarding the study design and provide some additional context.
Firstly, we would like to mention that the design of our study received funding from competitive public calls, which attests to its quality and potential impact. Furthermore, the manuscript underwent a rigorous review process by an expert committee, ensuring its scientific rigor and validity.
We would like to emphasize that the description of the musical intervention as well as the control group is detailed in the methods section, guaranteeing not only the reproducibility of the study but also providing crucial information for researchers in the field. By disseminating the results of our study, we aim to prevent other authors from duplicating efforts without adding novel contributions, thereby saving economic resources and human effort.
We have carefully considered the reviewer's suggestions and incorporated them into the limitations section, striving to present the study with clarity, ensuring that we are clear and transparent.
Moreover, it is important to note that the work has received highly positive evaluations from the other two reviewers, further supporting its scientific merit. We sincerely hope that the reviewer 3 will reconsider our manuscript for publication.
We remain available to address any further questions or concerns you may have.
Thank you and best regards,
Cristina
Round 2
Reviewer 3 Report
My concerns from the previous review were not fully addressed in the current manuscript. The potential adverse effect caused by the "control group treatment" is not mentioned.
Especially the result section should be revised significantly. In the current version, Table 2 provides the p-value for the changes between the intervention and control groups. In my opinion, two sets of p-value should be provided before and after the interventions from each group.
The authors need to convince the readers that the decline in the QOL-FV score for the control group is attributed to the natural course due to disease progression. It is hard to believe that only 7 days apart would have this apparent decline. Further references are needed to justify the explanations.
It is apparent that the 30-min basic therapeutic education is not a suitable reference for control. It is helpful to explain further about the alternatives and the reasoning behind using it.
In the limitation section, it is crucial to address the shortcoming in the research design. If the authors wish to "prevent other authors from duplicating efforts", it is important to help them to avoid making the same mistakes.
Due to the flaws in the research design, the result of the study might not show the true effect of music intervention, but the comparison between the music intervention and intervention of repetitive boring education. If the control is actually "no intervention", the effect of music may not exit. What is explained in the current manuscript is simply not compelling.
Author Response
Response to Reviewer 3, round 2 Comments
Point 1: My concerns from the previous review were not fully addressed in the current manuscript. The potential adverse effect caused by the "control group treatment" is not mentioned.
Response 1: This has been better specified according to the reviewer’s comments which are detailed below.
Point 2: Especially the result section should be revised significantly. In the current version, Table 2 provides the p-value for the changes between the intervention and control groups. In my opinion, two sets of p-value should be provided before and after the interventions from each group.
Response 2: Corrected after suggestion. Thanks.
Point 3: The authors need to convince the readers that the decline in the QOL-FV score for the control group is attributed to the natural course due to disease progression. It is hard to believe that only 7 days apart would have this apparent decline. Further references are needed to justify the explanations.
Response 3: We would like to convince the reviewer that our decline in the QOL-FV score is not strange. As we have mentioned, there is scarce previous research similar to our study, so it is complicated to show similar data. However, we have included, according the reviewer suggestion, another article that support our results:
- Article of Ito & Tadaka, 2022 (https://doi.org/10.1016/j.ctcp.2021.101508). They did a home-based palliative care trial with family caregivers of advanced cancer. They included a 7-day intervention consisted in an online diary program which aimed to enhance emotional competence. The results showed that the quality-of-life scores of family caregivers in the intervention group were maintained after the intervention, whereas those in the control group declined. Thus, they concluded that “The results suggest that the ONDIARY program in addition to usual care has potential to be effective in preventing decline and maintaining QOL of family caregivers of patients with advanced cancer in home-based palliative care settings.”
Point 4: It is apparent that the 30-min basic therapeutic education is not a suitable reference for control. It is helpful to explain further about the alternatives and the reasoning behind using it.
Response 4: We kindly disagree, however, this has been discussed in more depth according to the reviewer's suggestion in Limitations/Discussion. All the new changes are highlighted in yellow.
Point 5: In the limitation section, it is crucial to address the shortcoming in the research design. If the authors wish to "prevent other authors from duplicating efforts", it is important to help them to avoid making the same mistakes.
Response 5: Included after suggestion. Thank you very much. Although other studies with 7-day interventions using music with an impact on quality of life and emotional measures can be found as examples, DOIs:
- 1093/jmt/47.1.27
- 1111/aas.12100
- 10.1177/1533317509333258
- 1093/jmt/44.2.113
Also protocols: 10.1186/s13063-022-06448-w
Point 6: Due to the flaws in the research design, the result of the study might not show the true effect of music intervention, but the comparison between the music intervention and intervention of repetitive boring education. If the control is actually "no intervention", the effect of music may not exit. What is explained in the current manuscript is simply not compelling.
Response 6: We disagree. The reviewer considers the control group as “repetitive and boring” and we have not preceived this. As it has been detailed in the manuscript, each 30-min session was different of the previous one. The caregivers did not tell us that it was boring. They were able to learn more thinking that they were in the intervention group, consequenlty avoiding the placebo effect.
We received competitive funding, if we were include the control group as “no intervention”, our design would be rejected because we could not concluded whether the music intervention is effecttive or whether the fact that known that you receive more daily care produces the improvements.
We kindly ask the reviewer to consider our study again. The reviewer is highly contributing to improve our manuscript and, under our opinion, the revised version is better thant the previous one and the limitations are more completed. Nevertheless, our study can also contribute to the scientific literature.
